# Predicting the Glycemic Index of Biscuits Using Static In Vitro Digestion Protocols

**DOI:** 10.3390/foods12020404

**Published:** 2023-01-14

**Authors:** Xingguang Peng, Hongsheng Liu, Xuying Li, Huaibin Wang, Kejia Zhang, Shuangqi Li, Xianyang Bao, Wei Zou, Wenwen Yu

**Affiliations:** 1School of Food Science and Engineering, South China University of Technology, Guangzhou 510000, China; 2College of Agriculture and Biology, Shanghai Jiaotong University, Shanghai 200240, China; 3Department of Food Science & Engineering, Jinan University, Huangpu West Avenue 601, Guangzhou 510632, China; 4Longping Agricultural Science and Technology Huangpu Research Institute, Guangzhou 510700, China; 5Guangzhou Fine Nutrition Research Center, Guangzhou 510700, China; 6John A. Paulson School of Engineering and Applied Sciences, Harvard University, Cambridge, MA 02138, USA

**Keywords:** biscuits, in vitro digestion, INFOGEST, single enzyme, glycemic index

## Abstract

In vitro digestion methods that can accurately predict the estimated GI (eGI) values of complex carbohydrate foods, including biscuits, are worth exploring. In the current study, standard commercial biscuits with varied clinical GI values between 9~30 were digested using both the INFOGEST and single-enzyme digestion protocols. The digestion kinetic parameters were acquired through mathematical fitting by mathematical kinetics models. The results showed that compared with the INFOGEST protocol, the $AUR_{180}$ deduced from digesting using either porcine pancreatin or α-amylase showed the best potential in predicting the eGI values. Accordingly, mathematical equations were established based on the relations between the $AUR_{180}$ and the GI values. When digesting using porcine pancreatin, GI= 1.834 + 0.009 $\times AUCR_{180}$ ($R^{2}=$ 0.952), and when digesting using only α-amylase, GI= 6.101 + 0.009 $\times AUCR_{180}$ $(R^{2}=$0.902). The $AUR_{180}$ represents the area under the curve of the reducing-sugar content normalized to the total carbohydrates versus the digestion time in 180 min. The in vitro method presented enabled the rapid and accurate prediction of the eGI values of biscuits, and the validity of the formula was verified by another batch of biscuits with a known GI, and the error rate of most samples was less than 30%.

## 1. Introduction

For humans, the available daily energy provided by food carbohydrates is around 17 kJ/g (4 kcal/g), which accounts for 40~75% of the total energy intake (FAO/WHO, 1998) [1]. Carbohydrates in food (e.g., starch) are digested in the upper gastrointestinal tract into glucose to increase the human postprandial glycemic index (GI). Based on its influences on human blood glucose response after a meal, carbohydrates can be classified into glycemic and non-glycemic carbohydrates. Starch, the most abundant glycemic carbohydrate, is widely present in human staple foods and snacks (i.e., biscuits, cookies, potato crisps) [2,3]. Biscuits are the most typical ones usually consumed directly without cooking, providing rich nutrition and energy for humans [4,5]. As healthy eating takes root, consumers are tending to prefer biscuits with a lower postprandial GI.

While the chronically high postprandial blood glucose values caused by the rapid digestion of starch or starchy foods is usually associated with diseases such as type-2 diabetes (T2DM), obesity and insulin resistance (IR) [6,7,8,9,10], a low and stable postprandial blood glucose level helps to control appetite and delay hunger, and thus it could effectively prevent common non-communicable diseases (NCDs) including cardiovascular diseases, diabetes, obesity, bladder cancer, dyslipidemia and cancer [10,11,12,13,14,15].

The concept of the GI was first introduced by Jenkins et al. [16,17]. It provides the means of quantitatively comparing blood glucose responses after ingesting equivalent amounts of digestible carbohydrates from different foods. In 2010, the GI was well defined by the ISO (International Organization for Standardization) method 26642:2010 [18]. It is now generally accepted that foods that can be digested, absorbed and metabolized quickly are high-GI foods (GI > 70 on the glucose scale), whereas those with GI < 55 are low-GI foods. Nonetheless, the GI has been controversial since it was introduced, particularly because measurement of the GI is complicated and can be influenced by many factors. Additionally, the glycemic load (GL) is expressed as the product of the glycemic index of a food and its available carbohydrates, again reflecting the effect of the food itself on blood glucose.

Compared to clinical measurements where human subjects are needed, the prediction of the GI or GL using in vitro digestion protocols, from single static systems to multi-compartmental dynamic systems, has been studied as a potentially useful pathway as they are much cheaper, less time-consuming, have a better reproducibility, and importantly, they come with no ethic limitations [19,20,21,22]. To develop a more cost-effective GI prediction method, a common approach is to use artificial mechanical systems, which were developed together with the application of digestive enzymes to mimic the oral, gastric and small-intestinal digestive phases of in vivo digestion. Until now, static in vitro digestion experiments have shown great potential in predicting GI values. For example, Zou [23] reported that in rice powders with particle sizes ranging between 0~300 μm (Q-300), the in vitro digestibility was significantly and positively correlated with clinical GI values. Edwards [20] also found that with a fixed enzyme–starch substrate ratio, the kinetic parameters of the enzyme were most strongly correlated with in vivo rankings for the GI values of matched food products (*p* < 0.01).

Nonetheless, there are still non-negligible limitations that exist with current studies in terms of measuring the GI values in vitro, while GI testing still remains a priority. Moreover, it is also important to mention that even for the same type of cereal or cereal product, there are wide variations in GI values, presumably arising from variations in their manufacturing methods. For example, bread, breakfast cereals, rice and biscuit are all available in high-, medium-, and low-GI versions [24]. For example, the available methodologies contain several shortcomings such as: (1) lacking enough samples with known clinical GI values of the same type (2) the digestion experiments are usually randomly designed without providing repeatable experimental information (i.e., what digestive enzymes were used? Should oral mastication and gastric digestion be included or not?) and (3) different foods may need different predicted methods (i.e., ready-to-eat carbohydrate foods and foods that need to be cooked). It is thus particularly challenging to compare the predicted results of even the same product across different laboratories [23,24,25]. Except for single-enzyme digestion experiments, the INFOGEST protocol, which is based on an international consensus developed by the COST INFOGEST network [19], may also be potentially used to predict the GI values of food products.

Accordingly, in the current study, standard commercial biscuits with known clinical GI values were digested by using either the INFOGEST or single-enzyme protocols. The obtained digestograms were then fitted by mathematical kinetics models to obtain a few digestion parameters. Based on the linear relationship between the digestion kinetics parameters and the GI values, the effectiveness of the established equations was evaluated using other commercial biscuits with known clinical GI values.

By doing this, the current study aimed to (1) compare which of the static in vitro digestion protocols is more statically reliable in predicting the GI values of biscuits and (2) establish more general mathematical equations to predict the GI values of biscuits. It was hypothesized that the digestion parameters deduced from the mathematical fitting of the in vitro digestograms could be effectively used to predict the GI values of biscuits. This study provides a rapid and reliable high-output screening protocol for food companies to predict the GI values of biscuits in vitro.

## 2. Materials and Methods

### 2.1. Materials

Eight standard commercial biscuits (named Samples 1~8) with known clinical GI values (as announced by the company and labelled on the package) from 9~30 were purchased online from the same company in Europe. Clinical GI assays were conducted using the ISO method 26642:2010 [18].

Another six commercial biscuits (named Samples 9~14) with known clinical GI values measured by human subjects (as announced by the company and labelled on the package) were purchased domestically from different companies. The main raw materials of the analyzed biscuits were sorghum flour or wheat flour. Clinical GI assays were conducted by using either the ISO method 26642:2010 [18] or the recommended industrial standards for the clinical estimation of the glycemic index of foods in China (WS/T 652-2019). The latter was established based on the method of the International Standards Organization ISO [18]. Thus, these two in vivo methods for predicting the GI values can be considered comparable. The glucose load was calculated as the value equal to the GI value multiplied by the carbohydrate content (%).

In this study, Samples 1~8 were set as the experimental group as they were purchased from the same company so that the influences of specific factors such as the manufacturer and production process on their GI values could be avoided. Moreover, in order to eliminate the interference (composition, production process) caused by the same batch of biscuits and verify the universality of the experimental results, we specifically selected another batch of biscuits (Samples 9~14) with different ingredients from different manufacturers as the control group. All samples were subjected to the same in vitro digestion experiments and analysis. The control group was used to verify the accuracy of the GI prediction method obtained by the experimental group.

A-Amylase (AMY) from Amyloliquefaciens (E-BAASS, 145 U/mL), a 60 U D-glucose (GOPOD Format) assay kit and a total starch assay kit (AA/AMG) were purchased from Megazyme International (Irishtown, Bray, Co. Wicklow, Ireland).

Pancreatin from the porcine pancreas (P7545) and pepsin (P7000) was purchased from Sigma-Aldrich (St. Louis, MO, USA). P-hydroxybenzoic acid hydrazide (PAHBAH, H106356) with a molecular weight of 152.15 was purchased from Aladdin (Shanghai, China). All other chemicals, including absolute alcohol (≥99.7%), were all of reagent grade and used as received.

### 2.2. Sample Preparation

Biscuits (50 g) were weighted and ground into fine powders using a stainless-steel cereal grain mill (HK-86 M Guangzhou Xu Lang Machinery Equipment Co., Ltd., Guangzhou, Guangdong, China). All biscuit samples were kept in a sealed bag and stored in a refrigerator at 4 °C prior to further analysis.

### 2.3. Chemical Compositions

The moisture content was measured by drying the samples in a vacuum oven at 110 °C overnight and recording the weight loss in triplicates. The total starch content was measured using a Megazyme Total Starch Kit (K-TSTA-1107, Megazyme, Ireland) in triplicates. Before measuring, the samples were washed twice with 2 mL absolute ethanol to remove sugars that may have affected the measurement of the total starch [26]. The data obtained based on three replicates were consistent with the nutritional data supplied by the producers, including an impressive low starch content. A standardized assay was applied for measuring the total starch, with maize starch included in the kit applied as a referee. The total protein content was measured by an automatic Kjeldahl analyzer in duplicates. The total lipid content was obtained by ether extraction after acid hydrolysis in duplicates [27,28].

### 2.4. Content of Monosaccharides and Oligosaccharides in Biscuits

In order to eliminate the influences of original monosaccharides and oligosaccharides on the digestion results, the free monosaccharides and oligosaccharides contents of all the biscuits were measured using high-performance liquid chromatography (HPLC) according to a previous study [29].

In brief, the samples (0.1 g) were mixed with 30 mL distilled water to extract the sugars in the mixture and were then filtered using 0.2 µm filter paper. Standard mixtures containing glucose, maltose, fructose and sucrose were prepared in a range of 1~27 μg/mL in distilled water. The samples were analyzed on an Agilent 1100 HPLC with ELSD detection. The HPLC solvent was 75% acetonitrile in water with a 1 mL/min flow rate using an Alltech carbohydrate column with dimensions of 4.6 mm × 250 mm. The ELSD was set to a nitrogen flow rate of 2 mL/min (at 87 °C).

### 2.5. In Vitro Digestion

Both the INFOGEST protocol and the single-enzyme digestion model using either pure α-amylase (AMY, from *Amyloliquefaciens*) or porcine pancreatin were applied to conduct the digestion experiment as given in detail as follows [19,20].

#### 2.5.1. Digestion Using the INFOGEST Protocol

Simulated saliva fluid (SSF), simulated gastric fluid (SGF) and simulated intestinal fluid (SIF) were prepared according to the scheme and are shown in Table 1. Moreover, all the prepared solutions were diluted with deionized water to a constant volume of 1000 mL for standby. CaCL_2_ (H_2_O)_2_ was only added on the day of use with a total volume of 0.1 and 0.35 mL for every 100 mL of SGF and SIF solutions, respectively.

The procedure for the in vitro digestion experiments using the INFOGEST protocol was carried out based on the study of Brodkorb [19] with minor modifications and is shown in Table 2. For digestion by using the INFOGEST protocol, 5 g sample powders were directly used as the protocol requested.

After centrifugation, the supernatant (100 μL) was used to measure the reducing-sugar content using a PAHBAH assay [30], which was then transformed to the percentage of starch digested (%) using Equation (1):(1)$$\%\;\mathrm{Digested}=\Delta A_{\mathrm{Sample}}\times \frac{100\;\mu L\times 1\;\mathrm{mmol}/L}{\Delta A_{\mathrm{Maltose}\;\mathrm{standard}}}\times 342\times g\times {\mathrm{mol}}^{-1}\times 50\times 50\times \frac{100\%}{\mathrm{Sample}\;\mathrm{weight}}\times \frac{324}{342}$$

Here, the absorbance at each time interval is denoted as $\Delta A_{\mathrm{Sample}}$. The absorbance from the standard maltose solution is given as $\Delta A_{\mathrm{Maltose}\;\mathrm{standard}}$. The value 50 × 50 is the computational multiple from 200 μL aliquots to obtain a 50 mL reaction solution, and 324/342 is the transformation coefficient from starch (monomer unit: anhydroglucose) to maltose in terms of weight.

#### 2.5.2. Digestion Using Single Enzyme

As is well-documented, the static in vitro digestion model mimicking the intestinal digestion phase of food nutrients has also been frequently applied for conducting in vitro digestion experiments using a single enzyme of either pure α-amylase or pancreatin from the porcine pancreas [20,23]. For biscuits, the particle size (distributions) has a non-negligible effects on starch digestibility and even postprandial blood glucose concentrations [31,32,33,34]. For example, a positive correlation between the postprandial glycemic response and smaller disintegrated particles of rice (0~300 μm) has been found [33]. Accordingly, in the current study, before digestion using the single-enzyme digestion protocol of either pure α-amylase (AMY) or porcine pancreatin, all the biscuit samples were sieved through a 65-mesh sieve (325 μm). Then, only the samples with consistent particle sizes of up to 325 μm were collected and used for digestion so that the test could be completed within an appropriate time (normally up to 2–3 h), as samples with larger particle sizes, to some extent, may need much more experimental time to be digested completely [23,35].

The enzyme solutions were prepared as follows:

α-amylase enzyme solution: with a ratio of 1.2: 10 (*v*/*v*), α-amylase was mixed with a 0.2 M sodium acetate buffer (pH = 6.0) containing calcium chloride (200 mM), 0.02% sodium azide (*w*/*v*) and magnesium chloride (0.49 mM).

Pancreatin enzyme solution: pancreatin powder was weighed and mixed with 0.2 M sodium acetate buffer (pH = 6.0) containing calcium chloride (200 mM), 0.02% sodium azide (*w*/*v*) and magnesium chloride (0.49 mM) to a final concentration of 4 mg/mL. The mixture was thoroughly mixed and centrifuged at 10,000× *g* rpm for 10 min. The supernatant was collected and kept at 37 °C for further utilization.

Digestion using pure α-amylase: sieved biscuits (500 mg) were weighed, mixed with 10 mL distilled water and kept in a water bath (Hei-Tec, Heidolph Instruments GmbH & Co., Schwabach, Bayern, Germany) at 37 °C for 30 s with constant stirring at 300 × rpm using a magnetic stirrer. A total of 200 of μL supernatant was taken and added into a 2 mL centrifuge tube containing 800 μL 0.3 M sodium carbonate buffer to stop the enzyme reaction. Immediately following this, 5 mL of prepared pre-warmed α-amylase solutions were added to the mixture to start the digestion.

Digestion using porcine pancreatin: based on the preliminary results, the sieved biscuits (only 300 mg) were weighed and mixed with 10 mL distilled water. The mixture was mixed thoroughly and kept in a water bath (Hei-Tec) for 30 s with constant stirring at 300 × rpm using a magnetic stirrer. Following this, 200 μL of the mixture was taken and added into a 2 mL centrifuge tube containing 800 μL of 0.3 M sodium carbonate buffer to stop the reaction. Immediately after that, 5 mL pre-warmed pancreatin solution was added to start the digestion.

The whole digestion progress using either pure α-amylase or porcine pancreatin was carried out at 37 °C with constant shaking at 300 × rpm. Aliquots (200 μL) were taken at 5, 10, 15, 20, 30, 60, 90, 120, 180 and 240 min and added into a 2 mL centrifuge tube containing 800 μL of 0.3 M sodium carbonate buffer. The reducing-sugar content was measured using a PAHBAH assay, where maltose (0.1~1 mM) was used as the standard sugar [30,36]. The obtained reducing-sugar content was then transformed to the percentage of starch digested (%) using Equation (2):(2)$$\%\mathrm{Digested}=\Delta A_{\mathrm{Sample}}\times \frac{100\;\mu L\times 1\;\mathrm{mmol}/L}{\Delta A_{\mathrm{Maltose}\;\mathrm{standard}}}\times 342\times g\times {\mathrm{mol}}^{-1}\times 5\times 150\times \frac{100\%}{\mathrm{Sample}\;\mathrm{weight}}\times \frac{324}{342}$$

Here, the absorbance at each time interval is denoted as $\Delta A_{\mathrm{Sample}}$. The absorbance from the standard maltose solution is given as $\Delta A_{\mathrm{Maltose}\;\mathrm{standard}}$. The value 5 × 150 is the computational multiple from 200 μL aliquots to obtain an 18.0 mL reaction solution, and 324/342 is the transformation coefficient from starch (monomer unit anhydroglucose) to maltose in terms of weight. The final starch digestion curves were plotted as the percentage of starch digested (%) versus the digestion time (min).

### 2.6. Logarithm-of-Slope (LOS) Plot Analysis

As has been well-studied, when starch or starchy foods are digested with relatively high concentrations of (pancreatic) α-amylase and/or amyloglucosidase for a sufficient time period, the extent of starch digestion can be empirically fitted to a first-order kinetics equation [37,38,39,40] using following equations (see Equations (3) and (4)):(3)$$C_{t}=C_{0}+\left(C-C_{0}\right)\times \left(1-e^{kt}\right)$$
(4)$$\ln\frac{dC_{t}}{d_{t}}=\ln\left(C\times C_{0}\right)-kt$$

Here, $C_{0}$ is the digested amount of starch at *t* = 0 min, $C_{t}$ (%) is the percentage of starch digested at time *t* (min), $C_{\infty }$ (%) is the corresponding digested starch amount at infinite digestion time, representing a situation in which no more starch substrate is available for digestion, k ($\min^{-1}$) is the rate coefficient of starch digestion and $\frac{dC_{t}}{d_{t}}$ was obtained from the formula of ($C_{i+1}-C_{i})/\left(t_{i+1}-t_{i}\right)$ (*i* = 1, 2, 3…).

Accordingly, the in vitro digestograms of all the samples using either pure α-amylase or porcine pancreatin were first fitted using a logarithm-of-slope plot (LOS plot) to determine the total amount of digestible fractions using Equation (4).

### 2.7. Fitting to the First-Order Kinetics Models

Based on the LOS plot results, either a combination of parallel and sequential (CPS) kinetics models or a single first-order kinetics (SK) model, depending on the total number of digestible fractions visualized by the LOS plot results, was applied to fit the digestograms using the following equations, respectively [39,41]. For the CPS model, the following equation was applied:(5)$$C\left(t\right)=C_{0}+\left(C_{1\infty }\right)\times \left(1-e^{-k_{1}t}\right)+\mathrm{if}\;(t>t_{2\mathrm{start},\;\;}\;\left(\left(C_{2\infty }\right)\times \left(1-e^{-k_{2}\left(t-t_{2\mathrm{start}}\right)})\right),\;0\right)+\ldots$$

Here, $C_{0}$ is the amount of starch digested at time 0. $C_{1\infty }$ and $C_{2\infty }$ are the maximum amount of starch digested at an extended time for digestible fractions 1 and 2, referring to the quickly and slowly digestible starch fractions ($S_{F}$ and $S_{S}$, respectively). $k_{1}$ and $k_{2}$ are the corresponding rate constants for each digestible fraction, and $t_{2\mathrm{start}}$ is the initiation time for the digestion of $S_{S}$. Although the equation only assumes two different digestible fractions, more digestible fractions above ≥3 could be included in the equation depending on the number of digestible fractions identified by the LOS plot. For the SK model (single digestible fraction), the following equation was applied:(6)$$C\left(t\right)=C_{0}+\left(C-C_{0}\right)\left(1-e^{-kt}\right)$$

Here, $C_{0}$ is the amount of starch digested at time 0. $C_{\infty }$ is the maximum amount of starch digested at an extended time, and k is the corresponding overall rate constant for the whole digestion process.

Equation (3) was only designed for an ideal condition where a single uniform starchy component is digested, showing only one linear digestive phase in the LOS plots. However, real starchy foods (e.g., pasta, noodles, biscuits) often include multiple starchy substrates, showing much more than a single phase, as shown in the LOS plot. To address the more complex real situation, the comprehensive Equation (5) was applied to depict each digestion phase of all the starchy substrates contained in foods. Being a primary math tool, here, the phenomenological LOS plots based on Equation (3) can show several linear phases to estimate the possible quantities of starchy substrates of all kinds.

### 2.8. Data Analysis

Mean values and standard deviations (SD) were calculated by SPSS 26.0 (Statistical Graphics Corp., Princeton, NJ, USA). Two-tail tests were carried out to determine significant differences between the two different factors, and *p* ≤ 0.05 and *p* ≤ 0.01 were used as thresholds of significance and extreme significance, respectively. Significant difference analysis was carried out using ANOVA with Tukey’s pairwise comparisons at the *p* < 0.05 confidence level.

## 3. Results and Discussion

### 3.1. Chemical Compositions of Biscuits

The chemical compositions, including the total starch, crude protein and lipid contents of all the biscuits used, are listed in Table 3. It was observed that all the samples contained a relatively lower starch content, from 6.45 ± 1.38% to 11.65 ± 0.13%, which was significantly lower than normal biscuits, which generally contain 50~60% starch [42,43]. In the meantime, no significant correlations between the total starch content and GI values were found, suggesting that the original starch content was not the only determinant of the GI values, at least for the samples used in the current study. Generally, for most cereal foods, starch is the main glycemic carbohydrate, thus it is mainly the digestion of starch that increases the postprandial blood glucose level. For example, a close relationship between starch digestibility and GI values has been reported by Edwards et al. [20] and Zou et al. [23]. Accordingly, in these and many other studies, the starch content and its digestibility have been regarded as potentially useful factors for predicting the GI value, which is worth confirming. Nonetheless, in this study, our samples seemed to belong to a special category, as the total starch contents were low and close to each other. The calculated starch digestibility was also relatively low, as can be seen in Figure 1. Moreover, careful observation showed that the GI values of the samples with similar starch contents and digestibility were significantly different (e.g., the total starch content and digestibility of Samples 7 and 8 were similar, whereas their GI values differed greatly). Therefore, we believe that the starch content (digestibility) of the samples may have had no absolute relationship with their GI value.

Moreover, all the biscuits contained a relatively higher total protein and lipid content, varying from 9.81 ± 0.41% to 16.78 ± 1.02% and from 35.83 ± 1.02% to 43.24 ± 0.98%, respectively, especially when the fat content of similar products is usually 20% [42,43]. A significantly positive correlation between the GI and GL values was observed (R^2^ = 0.933, *p* < 0.001), which was understandable as the GL value was calculated based on the GI values [24,44]. Lastly, the total free sugars of all the samples varied from 15.81 ± 0.02% to 21.27 ± 0.03%. Detailed sample information of all the rest of the samples is shown in Appendix A.

For the test samples, we specifically selected some biscuits with significantly different compositions to verify the effectiveness of our model, the differences being mainly in terms of the content of total starch, fat and protein (as shown in Appendix A). For example, the protein content of Samples 9, 10 and 11 was smaller compared to the experimental group (Samples 1–8), while the protein content of Samples 12, 13 and 14 was larger than that of the experimental group. In addition, the fat content of all the test samples was significantly lower compared to the experimental group, while the carbohydrate and total starch contents were larger. Similarly, there was no strong correlation between the total starch content and the GI value of the two groups of samples.

The GI values of the analyzed biscuits were provided directly by the manufacturers and labelled on the package. The GL values were obtained by multiplying the GI values provided by the manufacturer by the carbohydrate content (%) of the samples. Accordingly, no standard deviations were provided for these two, as shown in Table 2.

### 3.2. In Vitro Digestion Results Using Different Models

Figure 1A shows the percentage of starch digested using the INFOGEST protocol. All the biscuits were digested similarly, among which the percentage of starch digested in Sample 3 (GI = 11) was the highest, followed by that in Sample 7 (GI = 16), whereas the total percentage of starch digested in Samples 4 and 5, both with a GI value of 13, were the smallest. This suggested that for the biscuits, the percentage of starch digested was not strictly correlated with their GI value, that is, a larger GI did not necessarily correlate with a higher content of starch digested. In this regard, this thus challenges the current in vitro measurement of food GI values when using starch amylolytic enzymes.

Figure 1B shows that when the samples were digested using only porcine pancreatin, the digestogram of Sample 8 (GI = 30) was completely different from the rest of the samples, which was probably attributed to its high GI value. In the meantime, similarly to the digestion results obtained by using the INFOGEST protocol, with the exception of the Sample 8, the percentage of starch digested in Sample 3 was also higher than that of the rest of the samples, followed by Sample 7. In contrast, the total percentage of starch digested in Sample 4 was the lowest.

Moreover, as shown in Figure 1C, when digesting the biscuits by using pure α-amylase, it was interesting to note that the total percentage of starch digested in Sample 2 (GI = 9) was very similar to that of Sample 7 (GI = 16), again suggesting that the GI values of the biscuits were not strictly controlled by the total percentage of starch digested. The difference in results (LOS plots (A, D) vs. (B, E) vs. (C, F)), as shown in Figure 2, were ascribed to the three distinctive enzymatical digestion systems, and we think this may have been caused by the type of enzyme or the gastric digestion stage. This showed that there were significant differences between the different digestive protocols, which definitely needs further study.

As shown in Figure 1, significant differences were observed for Samples 9~14, even for the same samples when digested by different protocols. For example, when digested by pure α-amylase, the total percentage of starch digested was significantly higher than those digested by either porcine pancreatin or by using the INFOGEST protocol, which was probably attributed to factors such as different enzyme activities and/or the particle size. Importantly, as mentioned above, for the digestion experiments using the INFOGEST protocol, although all the biscuit samples were ground into powders under the same conditions, unlike those digesting by using a single enzyme, they were not sieved and were directly used for the digestion experiments. In this respect, it is guaranteed that the particle size may have had an essentially important influence on the digestibility, and thus this made the results using different protocols incomparable. However, in the current study, we strongly insist that there was a completely no need to sieve the samples when digesting by using the INFOGEST protocol, as firstly, the INFOGEST protocol has been standardized and secondly, we only aimed to compare the suitability of the different in vitro protocols in predicting the GI values of the biscuits. In this regard, the experimental results shown in Figure 1 and elsewhere were still comparable as they were all digested using the standardized protocol, even though the mechanism for the variation in the results may have varied from one to another.

In addition, the six samples in the control group had little difference in terms of digestion using the INFOGEST system, but they had a slight difference using the simple single-enzyme system. As shown in Figure 1, Samples 11 and 12 had larger digestion rates and digestibility. This also showed that the GI value was independent of starch digestibility for our samples. We think the standardized procedure of INFOGEST may have made the samples behave more similarly. For example, after two hours of gastric digestion, most of the protein in the samples is hydrolyzed. The single-enzyme system may have retained the differences between the samples to obtain the different digestion results.

### 3.3. LOS Plot Analysis of Starch In Vitro Digestograms

The LOS plots of all the samples can be seen in detail in the Appendix A, while a typical example is shown in the main text. As shown clearly in Figure 2 and Appendix A, regardless of digesting using either the INFOGEST or single-enzyme protocols, the linear continuity of the LOS plot indicated that only one digestible phase existed, suggesting that the digestion of the biscuits was homogeneous. Figure 2A–C shows the starch digestibility of sample 1, and D–F shows the content of reducing sugar, the trend of both is consistent. The LOS diagram of Samples 9~14 used for validation was also analyzed. Appendix A shows that all the samples had only one digestion stage. Following the SK model, the starch digestibility could be calculated by Equations (3) and (4). This was significantly different from the digestive characteristics of other raw/uncooked starches (i.e., raw barley flour and cooked pasta [36,40]), which usually contain two digestible fractions.

Accordingly, based on the LOS plot analysis, the SK model was applied to fit the digestograms, as reported previously [39,45]. The fitting parameters are shown in Figure 3. While $C_{\infty }$ represents the percentage of starch digested at the end of digestion where no more starch substrates are available, *k* represents the digestion rate. As shown in Figure 3A,B, not surprisingly, for all the samples, both k and $C_{\infty }$ showed significant differences, even for the same samples digested by using different protocols (the INFOGEST or single-enzyme protocols). For example, although Samples 4, 5 and 6 all had the same GI value of 13 (Table 3), they had very different starch digestion rates, and the digestion rate of Sample 4 was significantly faster than the other two samples.

Moreover, it was also observed that even for the same sample, when they were digested by using different static in vitro models, they showed very different digestion characteristics, and thus resulted in different kinetic parameters. For example, the k values of Samples 1, 2 and 3 showed no significant differences when digested by using either the INFOGEST protocol or by using pure α-amylase, both of which were significantly larger than that when digested by using only porcine pancreatin. It was speculated that this was probably because of the different enzyme activities and other factors such as the particle size and/or the digestion of protein by pepsin when using the INFOGEST protocol (Table 2). The inconsistent results among the three in vitro digestive models were firstly ascribed to the inherent variable systematic difference, while there might also be other complex potential factors. Typically, food protein or/and its hydrolysates have been widely found to slow the digestibility of starch [46,47,48], as in the results via the two single-enzyme methods (containing only pure α-amylase without gastric pepsin) used in this study, where samples 1, 2 and 8, with their lower protein content, had higher digestion rates. On the other hand, the pre-digestion of protein components via INFOGEST gastric pepsin hydrolysis beforehand showed a higher intestinal digestibility for Samples 4, 5 and 6. However, it was noted that similar results were not seen for Samples 1, 2 and 3 under the same INFOGEST digestion, suggesting the presence of other unknown important factors besides protein, which deserves a more comprehensive study in the future.

The digestion parameters of the six samples used for validation are shown in Appendix A. The digestibility of the samples in the different models was different. In all the in vitro digestion systems, the *k* values of Samples 9, 10 and 11 were higher than those of the other three samples. This was because Samples 9, 10 and 11 contained higher total starch and lower protein contents. The *k* values of Samples 12, 13 and 14 were close because their compositions were similar. In addition, the comparison of starch digestibility is shown in Figure 4B and Appendix AB. The digestibility of the α-amylase system was the highest, which may have been caused by the different digestion procedures and enzyme types.

### 3.4. Pearson’s Correlations between Experimental Data and Clinical GI Values

Table 4 shows the Pearson correlations among the in vitro digestion parameters and the GI or GL values of the biscuits (*n* = 8). When digested by using the INFOGEST protocol, either calculated based on the percentage of starch digested or on the released reducing-sugar content normalized to total carbohydrates, no significant correlations between the digestion kinetic parameters and the GI or GL values were found. This suggested that although the INFOGEST protocol is regarded as one of the most effective static in vitro digestion models, it, however, was probably not suitable for predicting the GI values of the carbohydrate foods (i.e., biscuits) used in this study. As shown in Table 4 as well, when calculated based on the total percentage of starch digested, only when using porcine pancreatin were significant correlations between the digestion kinetics parameters and the GI or GL values observed. For example, when the samples were digested by porcine pancreatin, both the $AUC_{90}$ and k values were significantly and positively correlated with the GI and GL values. Nonetheless, when calculated based on the reducing-sugar content normalized to total carbohydrates, the results showed that either when digesting by porcine pancreatin or pure α-amylase, significant correlations between the digestion kinetics fitting parameters and the GI or GL values were observed, especially with the area under the curve (AUC) within a different digestion time period (say, $AUCR_{90}$).

Table 4 shows that, compared with other parameters, for either digesting by porcine pancreatin or pure α-amylase, the $AUC_{180}$ values were both significantly and positively correlated with the GI (r = 0.979, *p* < 0.01 and r = 0.957, *p* < 0.01, respectively) and GL values (r = 0.976, *p* < 0.01 and r = 0.953, *p* < 0.01, respectively). This result was similar to a previous study, where the author reported that out of all the indices tested, the value of $C_{90}$, representing the extent of starch digested after 90 min, was the most strongly correlated with in vivo rankings for the GI values of matched food products [20]. Accordingly, the parameters of $AUC_{180}$ were applied to establish mathematical equations to predict the estimated GI (eGI) values of the other commercial biscuits, which can be displayed as follows:

(1)When digested using only porcine pancreatin:



(7)
$$GI=1.834+0.009\times AUCR_{180}\cdot R^{2}=0.952\;$$



(2)When digested using pure α-amylase:



(8)
$$GI=6.101+0.009\times AUCR_{180}\cdot R^{2}=0.902\;$$



Here, $AUCR_{180}$ represents the area under the curve (AUC) for digesting for 180 min and normalized to the total carbohydrate weight.

### 3.5. Digestion Results of the Other Six Commercial Biscuits

In order to evaluate the effectiveness of the established equation for predicting the GI values of the other commercial biscuits, we tested several other commercial biscuits with known clinical GI values. For all these biscuits, the in vitro digestion experiments were conducted using exactly the same conditions as the standards. As shown in Figure 4, significant differences were observed even for the same samples digested by different protocols. For example, when digested by pure α-amylase, the total percentage of starch digested was significantly higher than those digested by either porcine pancreatin or by using the INFOGEST protocol, which was probably mainly attributed to the different factors mentioned above. In addition, the six samples of the control group had little differences in when digested of the INFOGEST system, but they had a slight difference when digested by the simple single-enzyme system. As shown in Figure 4, Samples 11 and 12 had larger digestion rates and digestibility. This also showed that the GI value was independent of the starch digestibility for our sample. We think the standardized procedure of INFOGEST may have made the samples behave more similarly. For example, after two hours of gastric digestion, most of the protein in the samples was hydrolyzed. The single-enzyme system may have retained the differences between the samples to obtain the different digestion results. After fitting mathematically, the eGI values of these samples are shown in Table 4.

As mentioned, the in vitro digestograms were acquired and mathematically fitted to reduce into several parameters, as shown in Figure 5. The parameters such as *k* and $C_{\infty }$ were obtained and then used to calculate the AUCRS, from which it was observed that the AUCRS and its labelled GI values showed strong correlations, and thus the predicted GI values were calculated by substituting the formula we obtained, and the results were verified by comparing their residuals and are shown in Table 5.

As shown clearly in Table 5, while the established Equations (7) and (8) could be more effectively used to predict the GI values of the standard biscuits, they were less effective for the commercial biscuits with a higher error rate. For all the commercial biscuits, here, we proposed that the large prediction gap between the estimated GI (eGI) and the clinical GI values was probably attributed to the differences between the in vivo and in vitro methodologies and also probably the source of the manufacturers and the production process. For example, in this study, the first batch of samples came from the same manufacturer. In general, this was still reasonable and suggested that the biscuit type and its processing may have had a significant influence on the in vitro digestion results of the samples, and different digestion models may need to be chosen for different biscuits to obtain more accurate results. In conclusion, it was feasible to predict the food GI values from in vitro digestion, and the reducing-sugar content released during the in vitro digestion process was more reliable in predicting the GI values instead of the percentage of starch digested.

Although slightly different in structure or minor ingredients, biscuits of all kinds can be seen as traditional standard bakery snacks comprising mainly incompletely gelatinized starch (10~50%) with a relatively low moisture contents and a potentially wheat gluten networks. Based on the adequate biscuit standards with known clinical GI values, this study aimed to set up a reliable and representative mathematical linear relationship between the clinical GI and in vitro eGI values that must be calculated as accurately as possible according to the in vitro digestion parameters. It is crucial to select the most proper in vitro parameters, which are expected to be applied for all forms of biscuits with varied GI values and be user-friendly to follow up through a convenient simplified equation.

Accordingly, it must be noted that this is an overall trend between the GI and in vitro eGI values based on adequate quantities of biscuit standards, which indeed would not be altered by each biscuit sample with a specific digestibility due to individual differences in terms of structure or minor ingredients. The different eGI errors, as shown in Table 4, could only be attributed to the different uniformity of the biscuit samples, since more uniform samples would present a minor error. In contrast, less-consistent samples would create more uncertainty regarding more significant errors.

## 4. Conclusions

At present, their simplicity and low cost make in vitro digestion simulation tests a perfectly suitable method for predicting the eGI value of biscuits. In this study, we compared the accuracy of three different static in vitro digestion models in predicting the eGI values of commercial biscuits. It was found that compared with the complicated INFOGEST protocol, the single-enzyme model using either porcine pancreatin or pure commercial α-amylase presented a higher accuracy in predicting the eGI values. Additionally, the single-enzyme digestion protocol could reduce the experimental complexity and unnecessary interference. In addition, it is believed that the equivalent release of reducing sugar was a good reflection of the digestion process in vitro, and the area under the curve had a significant correlation with the GI value, which can be used to predict the eGI value of carbohydrate foods such as biscuits. Nonetheless, it should be kept in mind that the selection of the digestion model is related to the sample (i.e., cooked or uncooked), and an appropriate digestion model should be selected according to the specific situation of the tests. The work of this experiment is helpful for the large-scale screening of healthy carbohydrate foods with low GI values as well as the prediction of the eGI value of biscuits. It also proved the great potential of using the area under the equivalent-release curve of reducing sugar in predicting the real GI value more efficiently.

## Figures and Tables

**Figure 1 foods-12-00404-f001:**
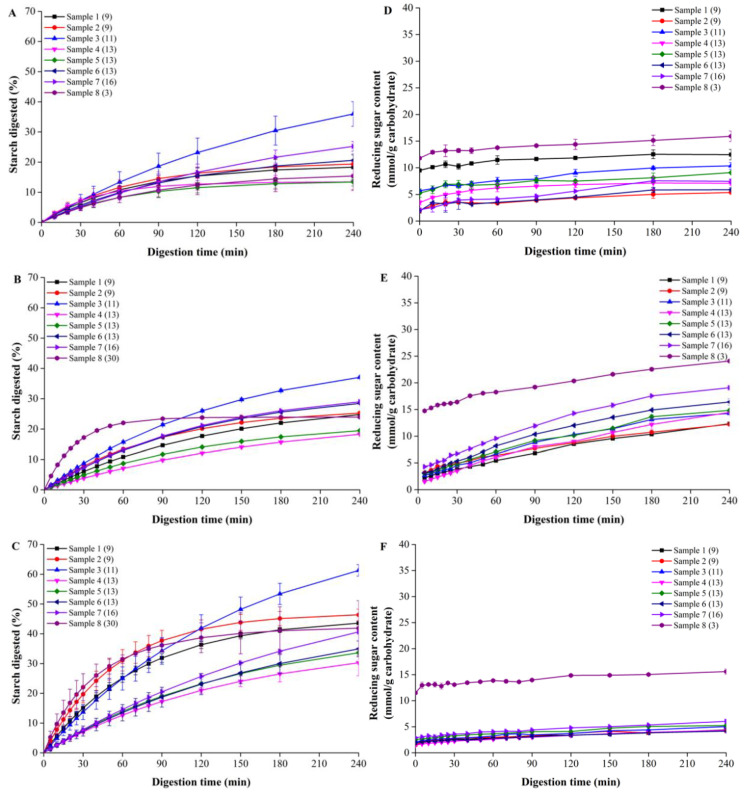
In vitro digestograms of all biscuits using different protocols, including INFOGEST (**A**,**D**) and single-enzyme protocol using either porcine pancreatin (**B**,**E**) or pure α-amylase (**C**,**F**). A, B and C are the digestion data of the percentage of starch digested. D, E and F are the released reducing-sugar content normalized to total available carbohydrates. The value in the brackets are the clinical GI values. Data is based on duplicate measurements.

**Figure 2 foods-12-00404-f002:**
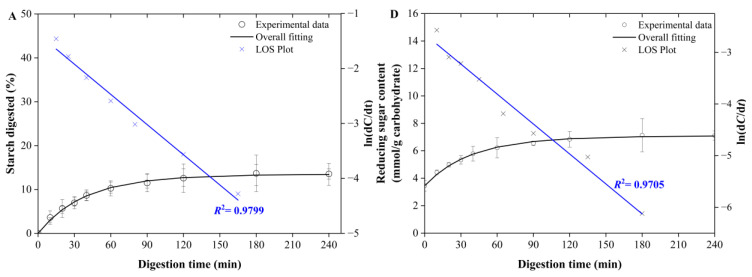
LOS plot for Sample 1 when digested using the INFOGEST protocol (**A**) and the single-enzyme digestion model using porcine pancreatin (**B**) or pure α-amylase (**C**), showing only one digestion phase. The left pane (**A**–**C**) contains the fitting results in terms of the digestion results of starch digested (%) as a function of the digestion time (min), while the right pane (**D**–**F**) contains the fitting results in terms of the digestion results of the total amount of released reducing-sugar content (mmol) normalized to the total carbohydrates as a function of the digestion time (min). Detailed sample information of all other samples is shown in Appendix A.

**Figure 3 foods-12-00404-f003:**
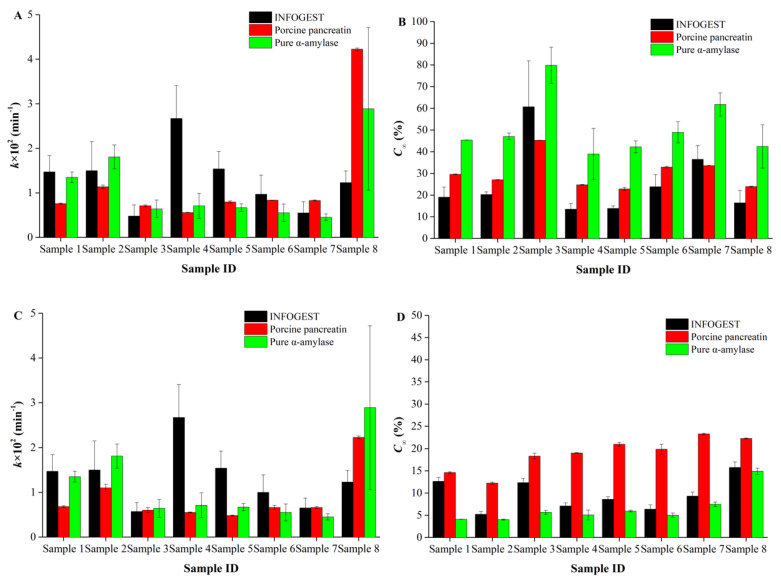
Fitting parameters of the in vitro digestograms using the single first-order kinetic (SK) model. (**A**,**B**) are the fitting results of the total percentage of starch digested; (**C**,**D**) are the fitting results of the reducing-sugar content, which was normalized to the total weight of carbohydrate. Data is based on duplicate measurements.

**Figure 4 foods-12-00404-f004:**
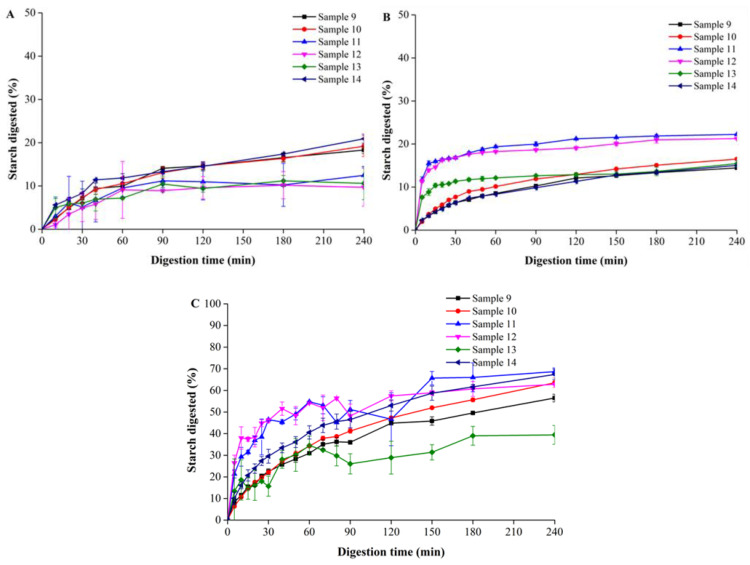
In vitro digestograms of all other commercial biscuits with different GI values using the INFOGEST (**A**) or single-enzyme digestion models by either porcine pancreatin (**B**) or pure α-amylase (**C**). Data is based on duplicate measurements.

**Figure 5 foods-12-00404-f005:**
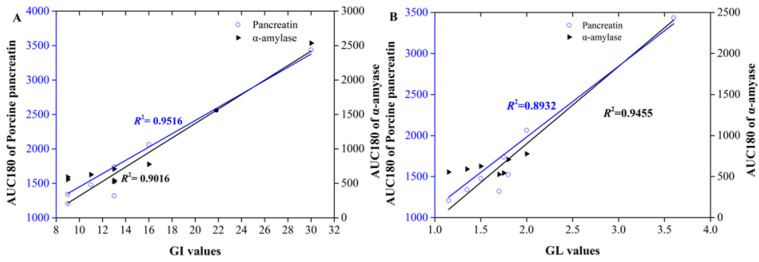
Linear regression analysis of the correlations between the $AUC_{180}$ and the GI (**A**) and GL (**B**) values when digesting using either porcine pancreatin or pure α-amylase (*n* = 8).

**Table 1 foods-12-00404-t001:** Chemical compositions of simulated solutions for in vitro digestion.

	g/L	M	SSF mL	SGF mL	SIF mL
KCL	37.3	0.5	30.2	13.8	13.6
KH_2_PO_4_	68	0.5	7.4	1.8	1.6
NaHCO_3_	84	1	13.6	25.0	85
NaCL	117	2		23.6	19.2
MgCL_2_ (H_2_O)_6_	30.5	0.15	1.0	0.8	2.2
(NH_4_)_2_CO_3_	48	0.5	0.12	1.0	-
HCL		6	0.18	2.6	1.4
CaCL_2_ (H_2_O)_2_	44.1	0.3			

**Table 2 foods-12-00404-t002:** In vitro digestion procedures of standard biscuits using INFOGEST protocol.

Procedures	Details
Oral	Biscuits samples (exactly 5.0000 g) g were weighed and mixed with 15 mL distilled water in a 125 mL Schott bottle with a screw cap. A 5 mL pre-warmed SSF solution was added into the mixture and immediately kept at 37 °C for exact 30 s in a water bath (WA20, Wiggens Gmbh, Straubenhardt, Germany) with constant stirring at 300 × rpm using a magnetic stirrer. No salivary α-amylase was added.
Gastric	Immediately following oral digestion, the food bolus was mixed with a pre-warmed SGF electrolyte stock solution to achieve of final ration of 1:1 (*v*/*v*). A 10 mL SGF solution containing 10 mg pepsin (1 mg/mL) was added. The mixture was kept at 37 °C for exact 120 min with constant stirring at 300 × rpm. The crucial aqueous acidity was adjusted in advance at pH ~1.8 for all the samples.
Intestinal	Immediately following gastric digestion, the pH of the mixture was adjusted to 7.0 using sodium hydroxide solution. A pre-warmed SIF solution (20 mL) containing pancreatin (4 mg/mL, *w*/*v*) was added to the gastric chyme to achieve a final ratio of 1:1 (*v*/*v*). The SIF solution containing 4 mg/mL of pancreatin needed to be centrifuged at 10,000× *g* for 10 min, and only the supernatant was used. The whole intestinal digestion was carried out at 37 °C with constant shaking at 300 × rpm for 120 min. Aliquots (200 μL) were taken at 0, 5, 10, 15, 20, 30, 60, 90 and 120 min time intervals and added into a 2 mL centrifuge tube containing an 800 μL 0.3 M sodium carbonate solution to stop the enzymatic reaction and were centrifuged at 10,000 × rpm for 10 min.

**Table 3 foods-12-00404-t003:** Chemical compositions and the standard GI and GL values of biscuits.

Sample ID	GI Values	GL Values	Moisture Content ^1^	Protein Content ^2^	Fat Content ^2^	Total Carbohydrates	Total Starch (%, db) ^1^	Free Sugar (%)
1	9	1.27	4.48 ± 1.09 ^d^	9.81 ± 0.41 ^c^	42.85 ± 1.74 ^ab^	14.15 ± 0.35 ^c^	7.33 ± 0.67 ^c^	21.57 ± 0.04 ^a^
2	9	2.00	4.59 ± 0.50 ^cd^	10.91 ± 0.57 ^c^	43.24 ± 0.98 ^a^	22.24 ± 1.59 ^a^	8.10 ± 0.31 ^c^	21.27 ± 0.03 ^b^
3	11	2.44	5.42 ± 0.05 ^bc^	15.29 ± 0.66 ^ab^	39.56 ± 1.09 ^abc^	22.14 ± 1.07 ^a^	6.45 ± 1.38 ^d^	15.81 ± 0.02 ^f^
4	13	2.37	5.43 ± 0.29 ^bc^	16.09 ± 0.56 ^ab^	35.83 ± 1.02 ^c^	18.21 ± 0.67 ^b^	11.91 ± 0.67 ^a^	17.08 ± 0.02 ^c^
5	13	2.35	6.62 ± 0.14 ^a^	14.00 ± 0.74 ^b^	36.00 ± 1.17 ^c^	18.10 ± 0.98 ^b^	11.65 ± 0.13 ^a^	16.17 ± 0.04 ^e^
6	13	2.34	6.69 ± 0.37 ^a^	16.00 ± 0.91 ^ab^	39.00 ± 0.73 ^bc^	18.00 ± 0.92 ^b^	8.93 ± 0.34 ^bc^	16.24 ± 0.03 ^e^
7	16	2.98	5.23 ± 0.16 ^bcd^	16.78 ± 1.02 ^a^	39.17 ± 0.95 ^bc^	18.65 ± 0.89 ^b^	9.57 ± 1.13 ^b^	16.80 ± 0.01 ^d^
8	30	4.76	5.80 ± 0.19 ^b^	13.59 ± 0.62 ^b^	39.65 ± 1.08 ^abc^	15.86 ± 0.51 ^bc^	9.95 ± 0.25 ^b^	15.84 ± 0.02 ^f^

^1^ Data was based on triplicate measurements; ^2^ data was based on duplicate measurements. Values with different letters in the same column are significantly different at *p* < 0.05.

**Table 4 foods-12-00404-t004:** Pearson’s correlations among the in vitro digestion parameters and clinical GI and the calculated GL values of all biscuits (*n* = 8).

	Digestion Models	Kinetics Parameters	Correlations	
			GI	GL
Starch	INFOGEST	$AUC_{90}$	−0.626	−0.634
		$AUC_{120}$	−0.554	−0.563
		$AUC_{180}$	−0.433	−0.443
		k	−0.104	−0.1
		$C_{\infty }$	−0.191	−0.199
	Porcine pancreatin	$AUC_{90}$	0.753 *	0.751 *
		$AUC_{120}$	0.673	0.671
		$AUC_{180}$	0.496	0.492
		k	0.915 **	0.916 **
		$C_{\infty }$	−0.314	−0.33
	α-amylase	$AUC_{90}$	0.279	0.275
		$AUC_{120}$	0.224	0.22
		$AUC_{180}$	0.136	0.13
		k	0.627	0.626
		$C_{\infty }$	−0.203	−0.21
Reducing sugar	INFOGEST	$AUR_{90}$	0.574	0.535
		$AUR_{120}$	0.57	0.53
		$AUR_{180}$	0.578	0.536
		k	−0.113	−0.11
		$C_{\infty }$	0.601	0.555
	Porcine pancreatin	$AUR_{90}$	0.974 **	0.973 **
		$AUR_{120}$	0.978 **	0.976 **
		$AUR_{180}$	0.979 **	0.976 **
		k	0.832 *	0.833 *
		$C_{\infty }$	0.653	0.65
	α-amylase	$AUR_{90}$	0.955 **	0.951 **
		$AUR_{120}$	0.956 **	0.951 **
		$AUR_{180}$	0.957 **	0.953 **
		k	0.627	0.626
		$C_{\infty }$	0.987 **	0.982 **

** Correlation is significant at the 0.01 level. * Correlation is significant at the 0.05 level.

**Table 5 foods-12-00404-t005:** Equations fitted to simple single-enzyme models (pancreatin and α-amylase) and predicted GI values for all samples.

Sample ID	GI Values	Digestion Model	Equation	eGI	Error Rate	Digestion Model	Equation	eGI	Error Rate
1	13	Pancreatin	GI = 0.009$\times AUCR_{180}$ + 1.834 $R^{2}$ = 0.952 *p* = 0.000	9.49 ± 5.55	27.02%	α-amylase	GI = 0.009$\times AUCR_{180}$ + 6.101 $R^{2\;}$ = 0.902 *p* = 0.001	10.72 ± 6.30	17.52%
2	13			12.63 ± 5.39	2.87%			13.18 ± 6.19	1.37%
3	9			8.78 ± 5.61	2.47%			11.22 ± 6.27	24.72%
4	11			12.08 ± 5.41	9.85%			12.36 ± 6.21	12.35%
5	16			15.67 ± 5.38	2.05%			13.19 ± 6.19	17.55%
6	13			13.82 ± 5.37	6.28%			11.23 ± 6.27	13.61%
7	30			29.44 ± 6.98	1.85%			30.05 ± 8.20	0.16%
8	9			12.30 ± 5.40	36.67%			12.32 ± 6.22	36.87%
9	52			39.96 ± 9.26	23.15%			28.74 ± 7.92	44.74%
10	53			38.43 ± 8.90	27.48%			31.75 ± 8.60	40.09%
11	44			39.66 ± 9.19	9.87%			26.38 ± 7.44	40.05%
12	30			49.11 ± 11.55	63.70%			41.20 ± 11.09	37.33%
13	28			20.81 ± 5.70	25.68%			20.54 ± 6.54	26.65%
14	19			13.87 ± 5.37	27.01%			19.07 ± 6.39	0.37%

## Data Availability

Data is contained within the article. The data that support the findings of this study are available from the corresponding author, (W,Y), upon reasonable request.

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
