# Peer review of "Predicting the Glycemic Index of Biscuits Using Static In Vitro Digestion Protocols"

_foods, 2023, doi:10.3390/foods12020404_

Round 1

Reviewer 1 Report

This work aims to compare static in vitro digestion methods to predict glycemic index (GI) values of standard commercial biscuits, and to establish more general mathematical equations to estimate those GI values. In this work, “it was hypothesized that the digestion parameters deduced from mathematical fitting of the in vitro digestograms could be effectively used to predict the estimated GI values of biscuits”. However, this study does not shed new light on human digestion's complex processes. The authors assumed highly simplified models (i.e., first-order kinetics models) for estimating GI without justifying this assumption. The use of empirical equations for starch-based food matrices also does not advance the understanding of how and when the starch is digested and what factors control this process. In my opinion, only a much more profound study would justify the publication of this work. In addition, grammatical and other such errors severely distract the reviewer.

Some comments that deserve further consideration are described below:

Introduction

Line 32: The range must be revised (~40% ~ 75% of the total energy intake…).

Materials and methods

2.1. Materials

- All reagents required to apply the INFOGEST protocol must be described.

2.2. Sample preparation

- How were the samples stored previous to further assays? Conditions?

- What was the particle size distribution obtained after grounding? This issue is critical to interpreting the subsequent results.

2.5. In vitro digestion

- Change INFOGESTO by INFOGEST.

- Table 1: Change INFOGES by INFOGEST.

- Table 1- Gastric procedure: It is relevant to precise the pH values reached when testing this stage.

2.6.1. Digestion using single enzyme

Lines 164-167: Why were particle sizes in the range of 0-325 μm selected?

2.7. Logarithm of slope (LOS) plot analysis

- What is the phenomenological approach for using equation 3 in this study?

2.8. Fitting to the first-order kinetics models

- What is the phenomenological approach for using equation 5?

Results (and discussions????)

3.1. Chemical compositions of biscuits

Lines 249-251: How do the authors rationalize those factors affecting the relationship between the total starch content and GI values obtained here?

3.2. In vitro digestion results using different models

Again, how do the authors rationalize the relationships found between the percentage of starch digested and biscuits ‘GI values?

3.3. LOS plot analysis of starch in vitro digestograms

- Figure 2: Why was sample 4 selected to show the LOS plots?

- Again, how do the authors interpret the differences found among LOS plots (A, D) vs. (B, E) vs. (C, F)?

- Lines 317-319: It is interesting to understand what other factors are consistently related to these findings.

Author Response

Comment 1: Introduction

  • Line 32: The range must be revised (~40% ~ 75% of the total energy intake…).

Response: thanks. This has now been revised in Line 33.

  • Comment 2: 2.1. Materials

All reagents required to apply the INFOGEST protocol must be described.

Response: thanks. Detailed information in terms of the INFOGEST protocol has been added in Table 1 in Line 164 and Lines 158 ~ 164 as follows:

‘…The simulated saliva fluid (SSF), simulated gastric fluid (SGF) and simulated intestinal fluid (SIF) were prepared according to the scheme and was shown in Table 1. Moreover, all the prepared solutions were finally diluted with deionized water to a constant volume of total 1000 mL for standby. The CaCL2 (H2O)2 was only added on the day of use with a total volume of 0.1 and 0.35 mL, respectively for every 100 mL SGF and SIF solutions…’

Table 1. Chemical compositions of simulated solution for in vitro digestion.

g/L

M

SSF mL

SGF mL

SIF mL

KCL

37.3

0.5

30.2

13.8

13.6

KH2PO4

68

0.5

7.4

1.8

1.6

NaHCO3

84

1

13.6

25.0

85

NaCL

117

2

23.6

19.2

MgCL2 (H2O)6

30.5

0.15

1.0

0.8

2.2

(NH4)2CO3

48

0.5

0.12

1.0

-

HCL

6

0.18

2.6

1.4

CaCL2 (H2O)2

44.1

0.3

  • Comment 3: 2.2. Sample preparation

How were the samples stored previous to further assays? Conditions?

Response: thanks a lot. To be more clear and in accordance with the reviewer concerns, we have added a brief description on page 3 line131~132 as follows:

‘…The sieved ground biscuit powders were kept in a sealed bag and stored in the refrigerator at 4 ℃ future analysis…’

  • Comment 4: What was the particle size distribution obtained after grounding? This issue is critical to interpreting the subsequent results.

Response: thanks. The ground biscuit powders were sieved using a 65-mesh sieve (325 μm), and only those could pass the sieve were collected for digestion. Accordingly, the particle size of our samples used for the digestion experiment was in the range of 0 ~ 325 μm. The reason why we chose this particle size range was because that based on a similar previous study, the digestibility of rice powders with particle size ranging between 0 ~ 300 μm showed the strongest correlation with GI values. Similarly, in this study, we chose 325 μm. New text has been added in Lines 128 ~ 131 as follows:

‘…and then sieved through a 65-mesh sieve (325 μm). The biscuit powders with particle size between 0 ~ 325 μm were collected, and only used for digestion when using the single enzyme digestion protocol by either porcine pancreatin or pure α-amylase…’

  • Comment 5:v2.5. In vitro digestion

- Change INFOGESTO by INFOGEST.

Response: thanks. This has been revised.

- Table 1: Change INFOGES by INFOGEST.

Response: thanks. This has been revised.

- Table 1- Gastric procedure: It is relevant to precise the pH values reached when testing this stage.

Response: We are extremely grateful to reviewer for pointing out this problem. We apologize for the language problem in the manuscript. New text has been added in Table 2 as follows:

‘…The crucial aqueous acidity had been adjusted in advance at pH ~1.8 for all the samples…’

  • Comment 6: Lines 164-167: Why were particle sizes in the range of 0-325 μm selected?

Response: thanks, this is indeed a very good question. The reason why we choose the particle size of 325 μm was because that based on the study of Zou et al. (2020), of rice powders, the sum-ratio of disintegrated-particle 0 ~ 300 μm correlated positively with clinical GI values. In the meantime, larger size of particles, to some extent, may need much more experimental time to be digested completely by each single enzyme. Accordingly, we selected the powders ranging between 0~325 μm for the digestion experiments so that the text could be finished in an appropriate time (normally up to 2~3 h). New text has been added in Page 5 Lines 187-189 as follows:

‘…(0 ~ 300 μm) has been found (Ranawana et al., 2010). Similarly, in this study, biscuit powders with particle size ranging between 0~325 μm were used for the digestion experiments, which is also to make sure that the digestion experiment could be finished in an appropriate time (normally up to 2~3 h)...’

  • Comment 7: What is the phenomenological approach for using equation 3 in this study? What is the phenomenological approach for using equation 5?

Response: thanks. We are extremely grateful to reviewer for pointing out this problem. To make it more clear, new text has been added in page 7 Lines 259 ~ 267 as follows:

‘…Eqn (3) is only designed for an ideal condition where a single uniform starchy component is being digested, showing only one linear digestive phase in LOS plots. However, the real starchy foods (e.g., pasta, noodles, biscuits) often include multiple kinds of starchy substrates to show much more than single phase as shown in LOS plot. To address the more complex real situation, the comprehensive Eqn (5) is thereby applied to clearly depict each digestion phases of all the starchy substrates contained in foods. Being a preliminary math tool, here the phenomenological LOS plots based on Eqn (3) can show number of linear phases to estimate the possible quantities of the starchy substrates of all kinds…’

  • Comment 8: Lines 249-251: How do the authors rationalize those factors affecting the relationship between the total starch content and GI values obtained here? Again, how do the authors rationalize the relationships found between the percentage of starch digested and biscuits ‘GI values?

Response: thanks. We are very grateful to the reviewers for raising this issue. New text has been added in page 9 Lines 282 ~ 295 as follows:

‘…at least for samples used in current study. Generally, for most cereal foods, starch is the main glycemic-carbohydrate, and thus it is mainly the digestion of starch that increases the postprandial blood glucose level. For example, a close relationship between the starch digestibility and GI values has been reported by Edwards et al. (2019) and Zou et al. (2020). Accordingly, in these and many other studies, starch content and starch digestibility have been regarded as a potentially useful factor for predicting the GI value of carbohydrate foods, which is worth confirming. Nonetheless, in this study, our samples seem to belong to a special category as the total starch content was low and close to each other. The calculated starch digestibility is also relatively low, as will be seen in Fig. 1. Moreover, careful observation will show that the GI values of samples with similar starch content and digestibility are significantly different either (e.g., the total starch content and digestibility of samples 7 and 8 are similar, whereas the GI values differ greatly). Therefore, we believe that the starch content (digestibility) of the sample in this study has no absolute relationship with its GI value…’

  • Comment 9: Figure 2: Why was sample 4 selected to show the LOS plots? Again, how do the authors interpret the differences found among LOS plots (A, D) vs. (B, E) vs. (C, F)?

Response: thanks. Indeed, sample 4 was only used as an example showing the fitting results using the kinetic models, the fitting results of rest samples were provided in Fig. A1.

New text has been added in page 2 Lines 344 ~ 347 as follows:

‘…The left pane (A, B & C) was the fitting results for the digestion results of starch digested (%) as a function of the digestion time (min), and the right pane was the fitting results in terms of the digestion results of the total amount of released reducing sugar content (mmol) normalized to the total carbohydrates as a function of the digestion time (min)…’

  • Comment 10: Lines 317-319: It is interesting to understand what other factors are consistently related to these findings

Response: thanks. We deeply appreciate the reviewer’s suggestion. This is indeed an interesting and noteworthy phenomenon, which we think may be caused by the type of enzyme or the gastric digestion stage. These phenomena are indeed worth studying, but the main purpose of this paper is to find a suitable in vitro digestion model combined with digestion parameters to predict the GI value of biscuits. These interesting phenomena just indicate that there are significant differences between different digestion models, which of course needs to be further studied. Still, new text has been added in page 3 Lines 366 ~ 373 as follows:

‘…For examples, when digesting using the INFOGEST protocol, the digestion of protein during the gastric phase, to some extent, would significantly increase the enzyme accessibility of starch molecules during the intestinal digestion phases, thereby leading to a larger k value, as seen with Samples 4, 5 and 6. However, for some samples such as Samples 1, 2 and 3, the digestion of protein by pepsin during the gastric digestion phase when using the INFOGEST protocol showed no significant differences compared with those digested using only pure α-amylase, this thus suggests that the protein is not the only factor in determining the starch digestibility…’

Reviewer 2 Report

The manuscript entitled “Predicting the glycemic index of biscuits using static in vitro digestion protocols” compared the accuracy of 3 different in vitro methods for the determination of glycemic index of biscuit products. The authors also compared the glycemic index calculated from the digestograms of digested starch and that of reducing sugar released. Mathematical modelling was used to create the predicting equations. I recommend the authors addressed the following points.

Abstract

L16-17. The correct name is INFOGEST.

The abstract did not mention about the part that evaluated the effectiveness of the equations.

Introduction

L47. The word “glycemic index” can be abbreviated.

L58. The abbreviation “GL” should be mentioned in full.

L72. What us regional food?

Materials and Methods

L100-110. It was unclear how the 2 groups of biscuits, Sample 1 - 8 and Sample 9 -14, were different. The ingredient list/ composition of each sample should be given in supplementary material.

L144. The correct name is INFOGEST.

L148-150. The authors mentioned about particle size of ground biscuit for digestion using single enzyme. Did the authors also control the particle size of ground biscuits used for digestion by INFOGEST protocol? Why the particle size was only controlled for digestion using single enzyme?

L169. How the particle size of 0 micron was measured?    

L207. The determination methods for glycemic index and glycemic load should be given.

Results

L248-249. The sentence “It was observed that all samples contained a relatively lower starch content…” was unclear. To which sample/food the starch content was compared?

L251-252. The sentence “Moreover, all biscuits contain a relatively higher protein and lipid content…” was unclear. To which sample/food the protein and lipid content were compared?

L258. What is the number of replicate for GI and GL values in Table 2?

L298. Why LOS plot of Sample 4 was selected to include in the manuscript, as they were 8 samples in total in Fig S1? Is there any specific reason/ justification?

L326. Table 2 mentioned in this section is actually Table 3.

L360. What is AUCR1801?

L365. Table 3 mentioned in this section is actually Table 4.

L381. It was unclear why the equations in Table 4 were different from the equations 7 and 8. How were they generated? If so, what are the equations that the authors would recommend for predicting the GI of biscuits.

Appendix

L431-440. This is not relevant and should be removed.

Author Response

  1. Comment: L16-17 L47 L58 L144: Some minor points: Some typos, some table annotation errors some abbreviations are missing (GI), or do not appear the first time mentioned (GL).

Response: thanks. We apologize for the language problems in the manuscript. We have now revised the manuscript very carefully.

  1. Comment: The abstract did not mention about the part that evaluated the effectiveness of the equations.

Response: thanks. We deeply appreciate the reviewer’s suggestion. According to the reviewer’s comment, we have now briefly described the feasibility evaluation of the equation in the abstract as shown in page 1 Line 26 ~ 27 as follows:

‘…and the validity of the formula is verified by another batch of biscuits with known GI, and the error rate of most samples is less than 30%...’

  1. Comment: L72 The sentence is not clear: What us regional food?

Response: thanks. We are grateful for the suggestion. New text has been added in page 2 Lines 74~77 as follows:

‘…not to mention that even for the same type cereal or and cereal products, it has a wide variation in GI values, presumably arising from variations in manufacturing methods. For example, breads, breakfast cereals, rice, biscuit were all available in high-,medium-, and low-GI versions…’

  1. Comment: L100-110. It was unclear how the 2 groups of biscuits, Sample 1 - 8 and Sample 9 -14, were different. The ingredient list/ composition of each sample should be given in supplementary material.

Response: thanks. All biscuits are different in raw materials, ingredients and GI values. We divided the same batch into one group. Detailed information of all biscuit samples have now been added in the supplementary file Table A1.

  1. Comment: L148-150 Did the authors also control the particle size of ground biscuits used for digestion by INFOGEST protocol? Why the particle size was only controlled for digestion using single enzyme? How the particle size of 0 micron was measured?

Response: thanks. We are extremely grateful to reviewer for pointing out this problem. For the digestion experiments using the INFOGEST protocol, we have not controlled the particle size of the ground biscuits as the procedure has been standardized for the INFOGEST protocol. To make it more clear, new text has been added in Lines 128 ~ 132 as follows:

‘…and then sieved through a 65-mesh sieve (325 μm). The biscuit powders with particle size between 0 ~ 325 μm were collected, and only used for digestion when using the single enzyme digestion protocol by either porcine pancreatin or pure α-amylase. The sieved ground biscuit powders were kept in a sealed bag and stored in the refrigerator at 4 ℃ future analysis...’

  1. Comment: The determination methods for glycemic index and glycemic load should be given

Response: Thank you for the suggestion. New text has been added in Lines 110~113 and 115~ 117 as follows:

‘…Their clinical GI assays were conducted using either the method of International Standards Organization ISO (Standardization, 2010) or the Recommended Industrial Standards for the clinical estimation of glycemic index of foods in China (WS/T 652-2019)….’

‘…In the meantime, the glucose load was calculated as the value equals the GI value multiplied by carbohydrate content (%)…’

  1. Comment: L248-249 The sentence “It was observed that all samples contained a relatively lower starch content…” was unclear. To which sample/food the starch content was compared ?

L251-252 The sentence “Moreover, all biscuits contain a relatively higher protein and lipid content…” was unclear. To which sample/food the protein and lipid content were compared?

Response: thanks. To make it more clear, new text has been added in Lines 277~278 and Line 296 as follows:

‘…which was lower than normal biscuits which generally contains a starch content of 50%~60% (Delamare et al., 2020; Ng et al., 2017)…’

‘…Especially the fat content, which is usually 20% for similar products…’

  1. Comment: L258. What is the number of replicate for GI and GL values in Table 2?

Response: thanks. The GI values of all biscuits were measured directly using in vivo method recommended by the International Standards Organization ISO (Standardization, 2010). For this method, at least a total number of 12 healthy volunteers were employed and their blood glucose concentrations were measured after ingesting a suitable amount of biscuits containing 25 g or 50 g total available carbohydrates. In this study, all the clinical GI values of these biscuits were provided directly by the manufacturers and labelled on the package. As for the values of GL, it was the value equals the GI value multiplied by carbohydrate content (%). Accordingly, no standard deviations were provided (of course there is no necessary) for these two in Table 2.

  1. Comment: L298. Why LOS plot of Sample 1 was selected to include in the manuscript, as they were 8 samples in total in Fig S1? Is there any specific reason/ justification.

Response: Thank you for your careful review. Indeed, we only randomly selected the data of sample 1 from the eight samples. We could of course list all data in Fig, but it shall be too large and thus the LOS fitting results of all rest samples except Sample 1 were provided in the supplementary file Figure A1.

  1. Comment: L326. Table 2 mentioned in this section is actually Table 3. Table 3 mentioned in this section is actually Table 4

Response: Thank you for your careful review. These have been revised accordingly.

  1. Comment: L381 It was unclear why the equations in Table 4 were different from the equations 7 and 8. How were they generated? If so, what are the equations that the authors would recommend for predicting the GI of biscuits.

Response: We are extremely grateful to reviewer for pointing out this problem. We are very sorry for the trouble caused by our mistakes, and now we have very carefully revised this. There are now the same.

  1. Comment: L431-440. This is not relevant and should be removed.

Response: thanks. This has been removed.

Reviewer 3 Report

What was the main raw material to production of all commercial biscuits? Were the biscuits wheat, gluten free or other?

Using different raw materials could be influenced on the result of eGI and GI. If raw material used for production of biscuits was different  why as a results Authors give only one equation calculated for all samples (line 360 and 362). Isn't that too much of a simplification?

Why in the paper i can find chemical composition only biscuits sample no 1-8 (table 2)? Please complete the missing data.

How Authors explain so different level of "Error rate" (table 4). Wouldn't the different chemical composition of each biscuit tested give different equations (like in line 360 and 362) and different error rates? For every biscuits sample should be different equation.

The results showed in table 4 were not discussed in the text. It is possible that  "Table 3" in line 389 should be "Table 4". Please check this part of paper.

Author Response

  1. Comment: What was the main raw material to production of all commercial biscuits? Were the biscuits wheat, gluten free or other?

Response: thanks. The main raw materials of all biscuits are sorghum flour or wheat flour and not gluten-free. New text has been added in Lines 110~111 as follows:

‘…The biscuit is made of sorghum flour or wheat flour…’

  1. Comment: Using different raw materials could be influenced on the result of eGI and GI. If raw material used for production of biscuits was different why as a results Authors give only one equation calculated for all samples (line 360 and 362). Isn't that too much of a simplification?

Response: We are extremely grateful to reviewer for pointing out this problem, We think comment 2 and comment 4 are similar, so we reply to you in comment 4 uniformly, hoping to satisfy you.

  1. Comment: Why in the paper i can find chemical composition only biscuits sample no 1-8 (table 2)? Please complete the missing data.

Response: Thank you for your careful review. We will add complete composition information of all samples in the appendix Table A1.

  1. Comment: How Authors explain so different level of "Error rate" (table 4). Wouldn't the different chemical composition of each biscuit tested give different equations (like in line 360 and 362) and different error rates? For every biscuits sample should be different equation.

Response: We are extremely grateful to reviewer for pointing out this problem. New text has been added in Lines 458~474 as follows:

‘…Despite might be slightly different in structure or minor ingredients, biscuits of all kinds can be seen as a traditional standard bakery snack which comprises mainly of incompletely gelatinized starch (10% ~ 50%), while a relatively low moisture content and potentially wheat gluten network. On basis of adequate biscuit standards with known clinic GI values, the study is thereby aimed to set up a reliable and representative mathematical linear relationship between GI and in vitro eGI values that must be calculated as accurately as possible according to the in vitro digestion parameters. It is crucial to assess to select the most proper in vitro parameters, which are expected to be applied for all forms of biscuits with varied GI values and also be user-friendly to follow up through a convenient simplified equation.

Accordingly, it must be noted that this is an overall trend between GI and in vitro eGI values based on adequate quantities of biscuits standards, which surely would not be altered by each biscuit’s samples with specific digestibility due to the individual differences in structure or minor ingredients. The different eGI errors as shown in Table 4 can only be attributed to the different uniformity of the biscuit’s samples, since the more uniform samples would be of smaller error while those less uniform samples would create more uncertainty in term of bigger error…’

  1. Comment: The results showed in Table 4 were not discussed in the text. It is possible that "Table 3" in line 389 should be "Table 4". Please check this part of paper.

Response: Thank you for your careful review. They are modified accordingly.